# Photoplethysmography-Based Blood Pressure Monitoring Could Improve Patient Outcome during Anesthesia Induction

**DOI:** 10.3390/jpm12101571

**Published:** 2022-09-23

**Authors:** Yan Degiorgis, Martin Proença, Yassine Ghamri, Gregory Hofmann, Mathieu Lemay, Patrick Schoettker

**Affiliations:** 1Dpt. Anesthesiology, Lausanne University Hospital and University of Lausanne, 1011 Lausanne, Switzerland; 2Systems Division, Centre Suisse d’Electronique et de Microtechnique (CSEM), Swiss Center for Electronics and Microtechnology, 2002 Neuchâtel, Switzerland

**Keywords:** optical blood pressure, cuffless blood pressure, photoplethysmography

## Abstract

During anesthesia, noncritical patients are routinely monitored via noninvasive cuff-based blood pressure (BP) monitors. Due to the noncontinuous nature of the monitoring, the BP values of the patient remain unavailable between consecutive cuff measurements, carrying the risk of missing rapid and sudden variations in BP. We evaluated the added value of using a photoplethysmography (PPG)-based continuous BP measurement device in addition to the standard cuff-based monitoring in a cohort of 40 patients in comparison with the current approach, in which only intermittent cuff-based measurements are available. When using a three-minute cuff measurement interval, using the PPG-based BP measurement in addition to the cuff-based monitor reduced the error (mean ± SD) of systolic (SBP) and mean (MBP) BP from 2.6 ± 19.6 mmHg and 1.2 ± 13.2 mmHg to 0.5 ± 11.2 mmHg and 0.0 ± 8.1 mmHg, respectively. Error grid analysis was also used to assess the improvement in patient safety. The additional use of the PPG-based BP measurement reduced the amount of data falling into higher risk categories. For SBP, points falling in the significant-, moderate-, and low-risk categories decreased from 1.1%, 8.7%, and 19.3% to 0.0%, 2.3%, and 9.6%, respectively. Similar results were obtained for MBP. These results suggest that using a PPG-based BP monitor—in addition to the standard cuff-based monitor—can improve patient safety during anesthesia induction, with no additional sensor needed.

## 1. Introduction

The routine monitoring of patients during anesthesia relies on the use of an oscillometric brachial cuff to measure arterial blood pressure (BP) [1]. This technique is based on the intermittent measurement of mean arterial blood pressure and extrapolation of systolic and diastolic values through automated calculation. The interval of measurement is generally selected between one and five minutes during the induction of anesthesia, depending on patients’ specifics and anticipated needs to monitor BP changes [2]. Technically, the minimum interval is around one minute due to the time necessary for the measurement to be collected. As a result, a “blind spot” exists in-between consecutive cuff measurements, during which important BP-related information is missed, which potential harms the patient. Poor blood pressure control is responsible for complications, such as myocardial injury, stroke, acute kidney injury, or even death [3,4,5,6,7].

An arterial catheter allows for continuous BP me and is known as the gold standard technique. However, due to its invasiveness, it is associated with various complications (such as infections, pseudo-aneurysms, vessel occlusions, or necroses) [8,9,10,11,12] and is therefore reserved for more fragile patients or complex procedures necessitating beat-by-beat BP control. The comparison of optical and cuffless blood pressure measurements using data acquired through an automated cuff might benefit the patients without the potential harm described above.

Continuous techniques using digital cuffs have been proposed, but they require specific devices and are limited in their applications; additionally, they can sometimes be unpleasant for the awake patient and raise questions about their accuracy and precision compared with the invasive gold standard that remains available [13].

Over the last two decades, several techniques based on photoplethysmography (PPG) have opened new perspectives [14]. Most of these techniques are based on the measurement of a pulse transit time (PTT), defined as the delay necessary to allow the transit of a pulse wave leaving the left ventricle outflow tract through the aortic arch and to the peripheral arteries. It has been established that the variation in BP is inversely proportional to the PTT, the latter being directly related to the distensibility of the arterial wall [15,16]. Thus, this parameter, combined with a simple calibration via a brachial cuff, allows for the continuous PPG-based monitoring of BP. However, measuring a PTT requires identifying the precise timing of the left cardiac ejection, which is challenging noninvasively. It may be an advantage of simplicity to use the concept of pulse arrival time (PAT), in which the onset is given by the R-wave peak of the electrocardiogram (ECG) and is therefore easily identifiable with standard monitoring. This simplification of using the PAT as a surrogate of the PTT comes at the cost of including part of the pre-ejection period (PEP), which depends on the ventricular electromechanical delay and isovolumic contraction.
PAT = PTT + PEP

Adding PEP to PTT involves a slight, clinically acceptable bias; the arrival of the pulse wave is detected at the periphery using a PPG-based device, such as a pulse oximeter. In addition, various studies have demonstrated feasible collection of the necessary variables during anesthesia induction [17,18].

In our study, we aimed to demonstrate that PAT-based continuous blood pressure measurement is a possible and accessible solution that can improve anesthesia safety compared with the use of a brachial cuff.

## 2. Method

### 2.1. Clinical Study

#### 2.1.1. Authorizations

The study was approved by the local ethics committee (CER-VD n° 327/15) and was registered under the number NCT02651558 at www.clinicaltrials.gov on 11 January 2016. The informed consent of all enrolled patients was acquired. This sample of patients was collected by Ghamri et al. to study the determination of BP variations from the PPG waveform [19].

#### 2.1.2. Patients Recruitment

Forty patients aged ≥ 18 years undergoing elective surgery necessitating general anesthesia for ENT or neurosurgery with IBP monitoring were recruited. The recruitment took place in 2017 at Centre Hospitalier Universitaire Vaudois (CHUV), and exclusion criteria were patient refusal, inability to consent, and arterial disease leading to an AP difference (>15 mmHg on SAP or >10 mmHg on DAP) between both arms [19].

#### 2.1.3. Anesthesia and Signal Acquisition

On the day of surgery, diuretics, angiotensin II receptor blockers and angiotensin-converting enzyme inhibitor therapy were suspended.

The patients were monitored (ECG, brachial cuff NIBP, and PPG) and connected to a Philips IntelliVue MP50 monitor (Philips, Amsterdam, The Netherlands). AP measurement was provided by the insertion of a 20-gauge, 4.5 cm length arterial catheter (BD Flowswitch; Becton-Dickinson, Franklin Lakes, NJ, USA) in the left or right radial (38 patients) or femoral artery (2 patients). In the case of radial artery catheterization, a fingertip was placed on the contralateral hand for PPG data acquisition.

Continuous IBP was recorded at induction of general anesthesia for 9–19 min with ixTrend express software version 2.1.0 (ixellence GmbH, Wildau, Germany).

General anesthesia was induced by propofol (2–3 mg/kg), fentanyl (1–2 μg/kg) or continuous remifentanil infusion (0.1–0.5 μg/kg/min), and rocuronium (0.6 mg/kg). Maintenance of anesthesia was provided by continuous propofol infusion (6–12 mg/kg/h) and hemodynamic support by boluses of ephedrine (5–10 mg) or phenylephrine (50–100 μg) or continuous norepinephrine infusion (0.02–0.2 μg/kg/min) [19].

Demographic values such as sex, age, height, weight, American Society of Anesthesiologists (ASA) score, comorbidities, site of catheterization, hypertension medication, and the type of surgeries were recorded.

#### 2.1.4. Sample Size

We referred to the sample size estimation by Ghamri et al. [19], who collected the data used in this study. A minimal sample size of 40—accounting for possible dropouts—was found, and thus 40 patients were enrolled.

### 2.2. Data Processing and Analysis

All signals were postprocessed and analyzed using MATLAB version 2020b (The MathWorks, Inc., Natick, MA, USA). The raw pulse oximeter PPG signal and the arterial line signal were aligned in time through cross-correlation. The latter was acquired in synchronization with the ECG signal, making all three signals synchronous. The obvious artifacts in the raw arterial line signal were visually identified and excluded. At each heartbeat, the PAT was estimated as the delay between the R-wave peak of the ECG signal and the minimum of the second time derivative of the PPG signal. A 20 s moving window was used to refine the value of the PAT by calculating it as the average of all individual PAT values in the previous 20 s, with prior removal of outliers using the median absolute deviation method [20]. Windows where the standard deviation (SD) of the nonrejected individual PAT values was greater than 20 ms were considered unreliable and rejected. Following the moving averaging procedure, all PAT values were transformed to logarithmic form. The processing of the arterial line signal was also performed on a beat-by-beat basis by extracting a systolic (SBP), mean (MBP), and diastolic (DBP) blood pressure value from each heartbeat before using a 20 s moving averaging window—as conducted for the PAT—for a meaningful comparison.

### 2.3. PPG-Based BP Estimation

The transformation of the logarithmic PAT values to the estimated BP values was achieved by training a linear model in a leave-one-out manner for each patient. To that end, the first PAT and invasive BP value of each patient’s recording was subtracted from all the patient’s values as follows:$$\begin{array}{c} x\left(n\right)=\log\left[\mathrm{PAT}\left(n\right)\right]-\log\left[\mathrm{PAT}\left(0\right)\right],\\ y\left(n\right)={\mathrm{BP}}_{\mathrm{INV}}\left(n\right)-{\mathrm{BP}}_{\mathrm{INV}}\left(0\right). \end{array}$$

By applying the leave-one-out procedure, the model used for a given patient was trained on all other patients in the dataset, except on the patient themselves. The model was a simple linear model with no intercept, meaning that it consisted of a slope parameter α. This parameter was trained by fitting in the least square sense $\left(\alpha \cdot x\right)$ values onto their corresponding y values using Tukey’s bisquare function to reduce the influence of outliers. The estimated PPG-derived BP values (${\mathrm{BP}}_{\mathrm{PPG}}$) of a patient were then obtained by multiplying the patient’s x values by α and adding ${\mathrm{BP}}_{\mathrm{INV}}\left(0\right)$:$${\mathrm{BP}}_{\mathrm{PPG}}\left(n\right)=\alpha \cdot x\left(n\right)+{\mathrm{BP}}_{\mathrm{INV}}\left(0\right)$$

The addition of ${\mathrm{BP}}_{\mathrm{INV}}\left(0\right)$ can be seen as an initialization (or calibration) of ${\mathrm{BP}}_{\mathrm{PPG}}$. This procedure basically consists of providing the PPG-based BP estimate with an initial BP starting point at the beginning of the monitoring session. In practice, this calibration procedure is typically carried out using an oscillometric cuff measurement. In patients monitored by a cuff-based BP monitor, it is standard to perform BP measurements every one to five minutes in highly dynamic conditions such as the induction of anesthesia. Therefore, in such patients, the PPG-based BP estimate (${\mathrm{BP}}_{\mathrm{PPG}}$) is not only calibrated at the beginning of the monitoring session but periodically recalibrated at each new cuff measurement. To simulate this use case in the present study, we tested several periodic recalibration intervals, from 1 to 5 min, by periodically readjusting ${\mathrm{BP}}_{\mathrm{PPG}}$ to the value of ${\mathrm{BP}}_{\mathrm{INV}}$, as illustrated in Figure 1.

### 2.4. Statistical Analysis

The agreement between ${\mathrm{BP}}_{\mathrm{PPG}}$ and ${\mathrm{BP}}_{\mathrm{INV}}$ was evaluated through Bland-Altman analysis [21], by computing the cohort-wise bias (mean difference) and 95% confidence intervals (CI), derived from the SD of the BP differences between both methods. Although not formally applicable to cuffless BP devices, the recommendations of the ISO 81060-2 standard for noninvasive sphygmomanometers were followed to evaluate this agreement, i.e., by taking into account the variability of the reference (${\mathrm{BP}}_{\mathrm{INV}}$) in the calculation of the error [22]. Although widely used, Bland-Altman analysis does not provide clear information on the clinical relevance of the relation between the two methods compared, hence the additional use of error grid analysis as proposed by Saugel et al. [23]. The error grid analysis allows for evaluating the risk induced by a BP estimation error on the treatment (or absence thereof) received by the patient as a consequence of this error. For instance, overestimating an MBP of 140 mmHg by 20 mmHg (i.e., estimating it at 160 mmHg) does not have the same clinical consequence as overestimating an MBP of 40 mmHg at 60 mmHg. In their paper, the authors defined five risk zones (no, low, moderate, significant, and dangerous risk) based on the expertise of 25 specialists (mainly operating room anesthesiologists). We used error grid analysis to evaluate our results in terms of clinical relevance and calculated the percentage of measurements falling in each risk category. Moreover, the intrasubject (patient-wise) Pearson’s correlation coefficients between the noninvasive BP estimate and the invasive reference were assessed.

### 2.5. Added Value vs. a Cuff-Based Monitor

As pointed out by Mukkamala et al. [24], the correct interpretation of the results obtained from calibrated cuffless devices can be difficult. For instance, a correlation coefficient calculated at a cohort-wise level may be artificially high simply due to the large inter-subject range of BP in the data. Moreover, in studies with low BP variability, or where the cuffless device is regularly recalibrated by a cuff-based measurement, simply using the latest cuff-derived calibration value as an estimate of BP may be good enough, and the PPG may not provide any added value. Therefore, to verify and quantify the added value of the PPG-based BP estimate compared with that of a simple periodic BP measurement provided by an intermittent cuff-based BP monitor, we also evaluated the error that such intermittent monitoring would induce by constructing ${\mathrm{BP}}_{\mathrm{CUFF}}$, a beat-by-beat BP estimate obtained by propagating the latest cuff-derived calibration value (see Figure 1). Note that the BP values used for each recalibration were removed from the analysis, both for ${\mathrm{BP}}_{\mathrm{PPG}}$ and ${\mathrm{BP}}_{\mathrm{CUFF}}$ when evaluating the agreement with ${\mathrm{BP}}_{\mathrm{INV}}$, as they would have artificially added zero-error points to the results.

## 3. Results

We present the results of an analysis of 40 patients with a radial (*n* = 38) or femoral (*n* = 2) catheter. Demographics, biometric characteristics, average BP values, and variability of the patients are shown in Table 1. All signals were recorded for a median per-patient duration of 11.3 min ([Q1, Q3] = [10.2, 12.5] minutes). The median per-patient data rejection rate, that is, the percentage of PPG-derived BP estimates considered unreliable (e.g., due to motion artifacts) and automatically excluded, was 1.1% ([Q1, Q3] = [0.5, 3.9]%).

A total of 32,100 heartbeats and their corresponding PAT values were available, out of which 1024 (3.2%) were rejected prior to analysis due to high within-window variability. The 40 trained linear models (one per patient, trained using all other 39 patients) resulted in an average (±SD) value of the α parameter of −164.1 ± 2.3 mmHg for SBP, −96.3 ± 1.3 mmHg for MBP, and −53.9 ± 0.9 mmHg for DBP, with corresponding coefficients of variations (SD over average) of 1.4%, 1.4%, and 1.7%, respectively. For each tested recalibration interval (from one to five minutes), the resulting mean and SD of the error are provided in Table 2 for both ${\mathrm{BP}}_{\mathrm{PPG}}$ and ${\mathrm{BP}}_{\mathrm{CUFF}}$. For instance, when using a three-minute recalibration interval, ${\mathrm{BP}}_{\mathrm{PPG}}$ reduced the error over ${\mathrm{BP}}_{\mathrm{CUFF}}$ on SBP and MBP from 2.6 ± 19.6 mmHg and 1.2 ± 13.2 mmHg to 0.5 ± 11.2 mmHg and 0.0 ± 8.1 mmHg, respectively. Bland-Altman plots for SBP and MBP using a recalibration interval of two minutes are shown in Figure 2 and Figure 3 for ${\mathrm{BP}}_{\mathrm{PPG}}$ and ${\mathrm{BP}}_{\mathrm{CUFF}}$ and ${\mathrm{BP}}_{\mathrm{CUFF}}$, respectively.

The corresponding error grid analysis plots are shown in the same figures. The percentage of points falling in each risk category as a function of the recalibration interval is detailed in Figure 4. Finally, Table 3 shows the median (and first and third quartiles) of the per-patient Pearson’s correlation coefficient between each noninvasive BP estimate (${\mathrm{BP}}_{\mathrm{PPG}}$ and ${\mathrm{BP}}_{\mathrm{CUFF}}$) and the invasive reference (${\mathrm{BP}}_{\mathrm{INV}}$).

## 4. Discussion

Anesthesia can expose patients to large and rapid variations in blood pressure that a single brachial cuff, due to the intermittent nature of the measurement. In our study, we demonstrated that PAT-based continuous blood pressure measurement allowed identification of inaccessible time points of blood pressure values compared with the use of a brachial cuff.

Comparing our PPG-based approach (${\mathrm{BP}}_{\mathrm{PPG}}$) with the standard cuff-based approach (${\mathrm{BP}}_{\mathrm{CUFF}}$) in the error grid analysis, we were able to identify (Figure 4) that the probability of harmful events was higher when using a single brachial cuff. In the purely cuff-based approach, the result of one measurement was considered representative of the patient’s BP until the next measurement (a few minutes later) occurred, even if the BP had changed, and with potentially severe errors in some cases. We noted that the limitation of a technique using PAT instead of PTT is the slight imprecision inherent to the PEP. However, comparing it with an iterative brachial cuff measurement, this small inaccuracy seemed to be largely admissible, as the PPG provided an added value in precision, accuracy, and reduced risk. We note that no error grid was provided by Saugel et al. [23] for DBP due to its lesser importance in settings such as general anesthesia induction.

Using error grid analysis, Takashi Juri et al. recently highlighted that MBP monitored by a brachial cuff could lead to overtreating hypo- or hypertension [25] due to the slight discrepancies between the noninvasive and invasive values. In their study, a better correlation was found between the invasive BP measurements and the brachial cuff for SBP, suggesting that this parameter is more reliable endpoint for assuring safety. In our study, due to the absence of cuff measurements, ${\mathrm{BP}}_{\mathrm{CUFF}}$ was simulated by “freezing” invasively collected data of the mean and systolic BP for one to five minutes, thus avoiding this problem. However, because ${\mathrm{BP}}_{\mathrm{PPG}}$ relies on a cuff-based calibration in a real-case scenario, it is important to bear in mind that a brachial cuff calibration may propagate an error of measurement to the following values until the next calibration.

An automated pulse oximeter waveform analysis by the oBPM^®^ algorithm was shown to accurately track acute blood pressure changes in this population [19] with a median per-patient data rejection rate of 4% compared with the invasively acquired blood pressure. While the latter method was described as the “gold standard technique”, it carries specific limitations and the potential for injury. In our current study, with an analysis based on the same population and with blood pressure measurement taken with a noninvasive cuff based monitor, we were able to demonstrate access to continuous BP measurement. Our current results are part of our goal to continuously improve cuffless blood pressure measurement and represent a new analysis of our existing data using alternative techniques.

In a study published in 2021 about the evaluation of the accuracy of cuffless blood pressure measurement devices [24], Mukkamala et al. detailed how the reliability of cuffless devices evaluated in too-stable hemodynamic conditions can be overestimated, as simply propagating the latest calibration measurement would provide acceptable results in most cases, hence the need to evaluate cuffless devices in dynamic environments. We collected our data during the induction of anesthesia in order to take advantage of the blood pressure variations that may occur.

In our study, the range of error increased over time and required an iterative recalibration using a brachial cuff. Considering a reasonable 3 min delay of recalibration, we found an SD of the BP differences of 11.2 mmHg for SBP and 8.1 mmHg for MBP. The ISO81060-2 standard requires the SD of the differences to be no greater than 8 mmHg. It is important to note that this standard is designed for noncontinuous measurement using cuff-based devices in a static and quiet environment, which hardly fits the settings of our study.

The limitation of our results resides in how our data were collected during the induction of anesthesia and not during the whole surgical procedure. The use of certain vasoactive drugs or anesthetics could have created biases, but this remains to be determined.

In the future, it may be interesting to collect data through anesthesia in a wide range of situations and patients. Alternatively, the possibility of using the PAT-based BP estimate to automatically trigger cuff measurements at specific time points, e.g., when a large BP variation is observed, should be investigated. These may help improve the usefulness of the cuff-based BP measurements by performing them at the most critical moments of the anesthesia.

## 5. Conclusions

The PAT-based approach seems to be a feasible alternative to the existing cuff-based measure of BP. It requires no additional device and just the implementation of an algorithm based on usual monitoring data (pulse oximeter, ECG, and brachial cuff BP).

Using an automated and continuous PPG-based technique would be feasible and seems reliable enough to increase safety during anesthesia induction by avoiding the potentially dangerous “blind spot” inherent in using a single brachial cuff. It may provide inexpensive complementary data to add to the usual monitoring. Currently, all existing PPG-based techniques require calibration to be accurate; thus, a full replacement of the brachial cuff is not yet possible. BP measurement using PAT as proposed is not meant to replace the use of an arterial catheter but only to increase safety. Therefore, patients who are severely ill, unstable, or require continuous hemodynamic support with vasoactive drugs should remain monitored by invasive means.

## Figures and Tables

**Figure 1 jpm-12-01571-f001:**
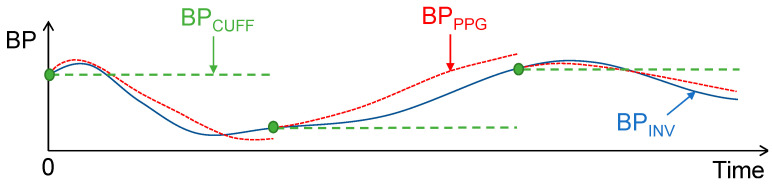
Calibration and simulation of error using an intermittent measurement device.

**Figure 2 jpm-12-01571-f002:**
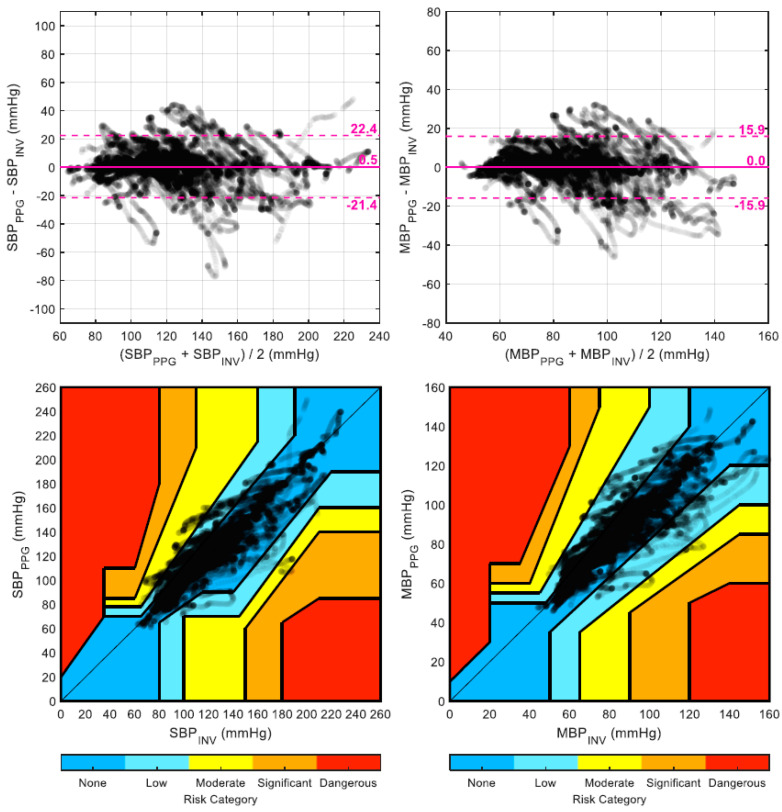
Bland-Altman plots and error grid analysis plots for BP_PPG_ (SBP/MBP).

**Figure 3 jpm-12-01571-f003:**
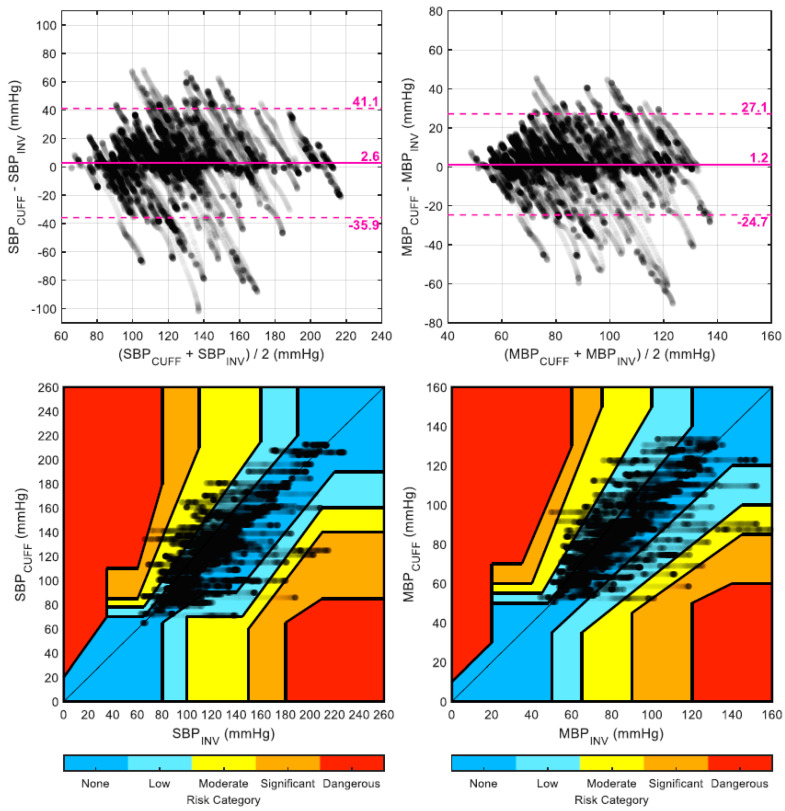
Bland-Altman plots and error grid analysis plots for BP_CUFF_ (SBP/MBP).

**Figure 4 jpm-12-01571-f004:**
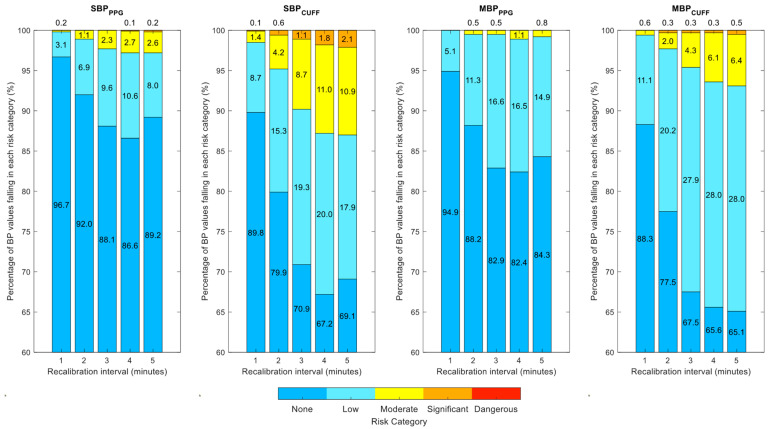
Percentage of points falling in each risk category as a function of the recalibration interval.

**Table 1 jpm-12-01571-t001:** Demographic and biometric characteristics of the 40 patients. The data are given as median (range) or count (percentage) over the cohort. The average blood pressure (BP) and variability values are the per-patient average value and per-patient minimal–maximal range of invasive BP throughout the entire recording. ENT: ear nose throat, ASA: American Society of Anesthesiologists, ACE: angiotensin-converting enzyme, ARB: ngiotensin II receptor blockers, SBP, DBP, and MBP: systolic, diastolic, and mean BP, respectively.

Patient Characteristics (*n* = 40)		Median (Range) or Count (Percentage)
Age (years)		62 (24–81)
Height (cm)		169 (154–195)
Weight (kg)		75 (46–118)
Body mass index (kg/m^2^)		25 (18–45)
Sex, male		22 (55.0)
Active smoking		16 (40.0)
ASA class	I	3 (7.5)
	II	23 (57.5)
	III	14 (35.0)
Type of surgery	ENT surgery	12 (30.0)
	Neurosurgery	20 (50.0)
	Spinal surgery	8 (20.0)
Comorbidities	Arterial hypertension	12 (30.0)
	Coronary artery disease	3 (7.5)
	Atrial fibrillation	3 (7.5)
	Arteriopathy	4 (10.0)
	Valvular heart disease	2 (5.0)
	Renal insufficiency	3 (7.5)
	Diabetes mellitus	6 (15.0)
	Dyslipidemia	9 (22.5)
Medication	Beta-blockers	6 (15.0)
	ACE inhibitors and ARBs	8 (20.0)
	Calcium channel blockers	3 (7.5)
SBP average (mmHg)		121 (83–200)
DBP average (mmHg)		63 (42–87)
MBP average (mmHg)		87 (58–121)
SBP variability (mmHg)		77 (29–134)
DBP variability (mmHg)		39 (19–79)
MBP variability (mmHg)		55 (24–103)

**Table 2 jpm-12-01571-t002:** Resulting mean and SD of the error for both BP_PPG_ and BP_CUFF_.

Mean ± SD of the Differences with BP_INV_ (mmHg)	Recalibration Interval				
	Every Minute	Every 2 Minutes	Every 3 Minutes	Every 4 Minutes	Every 5 Minutes
SBP_PPG_ − SBP_INV_	0.2 ± 6.0	0.4 ± 8.9	0.5 ± 11.2	0.5 ± 11.5	−0.4 ± 11.7
MBP_PPG_ − MBP_INV_	0.1 ± 4.4	0.2 ± 6.4	0.0 ± 8.1	0.2 ± 8.2	−0.7 ± 8.3
SBP_CUFF_ − SBP_INV_	0.8 ± 9.8	1.7 ± 13.3	2.6 ± 19.6	4.9 ± 20.4	3.8 ± 22.0
MBP_CUFF_ − MBP_INV_	0.4 ± 6.8	0.9 ± 10.3	1.2 ± 13.2	2.7 ± 13.5	1.5 ± 14.4

**Table 3 jpm-12-01571-t003:** Median (and first and third quartiles) of the per-patient Pearson’s correlation coefficient between each noninvasive BP estimate (BP_PPG_ and BP_CUFF_) and the invasive reference (BP_INV_).

Median (Q1, Q3) Patient-Wise Correlation Coefficient	Recalibration Interval				
	Every Minute	Every 2 Minutes	Every 3 Minutes	Every 4 Minutes	Every 5 Minutes
*ρ*(SBP_PPG_, SBP_INV_)	0.93 (0.87, 0.96)	0.88 (0.79, 0.94)	0.83 (0.69, 0.92)	0.88 (0.75, 0.93)	0.85 (0.72, 0.90)
*ρ*(MBP_PPG_, MBP_INV_)	0.89 (0.85, 0.94)	0.82 (0.74, 0.92)	0.80 (0.56, 0.89)	0.80 (0.66, 0.89)	0.79 (0.66, 0.88)
*ρ*(SBP_CUFF_, SBP_INV_)	0.83 (0.74, 0.87)	0.71 (0.45, 0.77)	0.48 (0.20, 0.67)	0.42 (0.19, 0.76)	0.36 (0.13, 0.63)
*ρ*(MBP_CUFF_, MBP_INV_)	0.82 (0.71, 0.86)	0.66 (0.36, 0.74)	0.37 (0.13, 0.66)	0.40 (0.10, 0.70)	0.37 (0.00, 0.60)

## Data Availability

The data presented in this study are available in tabulated form on request. The data are not publicly available due to ethical restrictions.

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
