# Peer review of "Photoplethysmography-Based Blood Pressure Monitoring Could Improve Patient Outcome during Anesthesia Induction"

_jpm, 2022, doi:10.3390/jpm12101571_

Round 1

Reviewer 1 Report

The authors have presented a nicely performed study using PPG to track changes in blood pressure between cuff measurements.  The inclusion of "last cuff pressure" as a comparator in the results is a nice touch to demonstrate the limitations of current practice and the potential utility of the use of PPG in this manner.

My only feedback for improvement is that the authors note that this data was collected as part of the study that collected the data for reference 19 (Ghamri et al 2020).  The authors should add a few additional sentences to more clearly explain how this work is new/novel compared to the original study they published on this topic, and whether this study was always planned as part of that work or if this was developed after completion of the first manuscript.

Author Response

Thank you for your pertinent remark. We added additional sentences in red in the introduction and discussion parts of the text to clarify.

Introduction: 

A comparison of optical and cuffless blood pressure measurements with data acquired through an automated cuff might benefit the patients without the potential harm described above.

Discussion:

An automated pulse oximeter waveform analysis by the oBPMTM algorithm has been shown to accurately track acute blood pressure changes in this population [19], with a median per-patient data rejection rate of 4%, as compared to the invasively acquired blood pressure. While this latter method has been described as the “gold standard technique”, it carries specific limitiations and potential of injury. In our current study, with an analysis based on the same population and with blood pressure measurement taken with a non-invasive cuff based monitor, we were able to demonstrate access to continuous BP measures. Our current results are part of our goal to continuously improve cufless blood pressure measurement and represent a new analysis of our existing data, using alternatove techniques.          

Reviewer 2 Report

Very good paper which can help reduce hypotension during anesthesia induction

2 comments:

1) the tables and graphs in the paper are not well placed ( table 1 in methods section, figure 1 in result)

2) The word : "patient safety" is maybe too strong....in the title, methods etc....can you find another word ? Because we don't have data showing that hypotension during anesthesia induction is deleterious ( prospective RCT), same for : preventing hypotension during anesthesia induction can decrease postop complications...

Author Response

Thank you very much for your pertinent remarks. We moved the table 1 to the correct place.

We also changed the title and replaced "patient safety" with "patient outcome
